# Vulnerability and Decision-Making in Multispecies Fisheries: A Risk Assessment of Bacalao (*Mycteroperca olfax*) and Related Species in the Galapagos' Handline Fishery

José F. Pontón-Cevallos [1,2,3,*], Stijn Bruneel [1], José R. Marín Jarrín [3,4,5], Jorge Ramírez-González [4], Jorge R. Bermúdez-Monsalve [3,6,7] and Peter L. M. Goethals [1]

1    Research Group Aquatic Ecology, Faculty of Bioscience Engineering, Ghent University, 9000 Ghent, Belgium; stijn.bruneel@ugent.be (S.B.); peter.goethals@ugent.be (P.L.M.G.)
2    Facultad de Ciencias de la Vida, Escuela Superior Politécnica del Litoral (ESPOL), Campus Gustavo Galindo, Guayaquil EC090151, Ecuador
3    Galapagos Marine Research and Exploration Program (GMaRE), Charles Darwin Foundation and Escuela Superior Politécnica del Litoral Research, Charles Darwin Research Station, Puerto Ayora EC200350, Ecuador; jose.marinjarrin@humboldt.edu (J.R.M.J.); jrbermud@espol.edu.ec (J.R.B.-M.)
4    Charles Darwin Foundation, Charles Darwin Research Station, Puerto Ayora EC200350, Ecuador; jorge.ramirez@fcdarwin.org.ec
5    Department of Fisheries Biology, Humboldt State University, Arcata, CA 95519, USA
6    Facultad de Ingeniería Marítima y Ciencias del Mar, Escuela Superior Politécnica del Litoral (ESPOL), Campus Gustavo Galindo, Guayaquil EC090151, Ecuador
7    Marine Environment Laboratories, International Atomic Energy Agency, 98000 Monaco, Monaco
*    Correspondence: josefernando.pontoncevallos@ugent.be

**Abstract:** Marine fish populations can be vulnerable to overfishing, as a response of their life history, ecology, and socio-economic aspects. Vulnerability assessments, in this regard, can be used to support fisheries decision-making by aiding species prioritization. Assessments like Productivity–Susceptibility Analyses are well suited for multispecies fisheries, with low gear selectivity and insufficient fishery-independent and dependent data. Using this method, we assessed local vulnerability of the Galapagos grouper ('bacalao'; *Mycteroperca olfax*) and compared it with other phylogenetically-related species caught in the Galapagos' handline-fishery. Bacalao is an overfished regionally endemic fish species, characterized by low resilience, high market and cultural value and high spatial overlap with the fishery. Our results suggested that bacalao is a species of high management priority, requiring urgent measures to prevent fisheries' collapse. In addition, if current fishing pressure persists, other related species may become threatened in the near future. We also evaluated different management scenarios using this approach. Results suggested that the inclusion of additional no-take zones in the marine reserve, comprising key nursery habitats (such as mangroves) and spawning aggregation sites, would be necessary to reduce species vulnerability and to benefit other related species. Improving enforcement and fishers' compliance are essential to guarantee the effectiveness of these measures.

**Keywords:** vulnerability; decision-making; multispecies fisheries; groupers; productivity susceptibility analysis; *Mycteroperca olfax*

## 1. Introduction

### 1.1. Fishing as a Threat for Marine Fish Species and Its Importance to Society

Humans throughout history have relied upon the oceans for the provision of goods and services [1]. Unfortunately, over a third of the world ocean area has already been moderately affected from the impacts of human activities (e.g., global climate change, pollution, habitat loss and degradation, species invasions, disease, and overexploitation of resources; [1,2]), and by the effects of environmental variability (e.g., hydrographic variability, climatic cycles; [3]).

Fishes are the most threatened group among vertebrates (combining freshwater and marine species), and many species have reduced substantially in their population sizes or become economically extinct [2,4]. Exploitation (by fishing) is considered the main cause for these declines or extirpations, as most of the world's marine fisheries are fully exploited or overexploited [5,6]. Fishing affects fish populations through direct mortality on target species, but also through indirect effects such as by-catch, habitat destruction, functional alterations of ecosystems, and human-induced evolutionary shifts in populations [7,8]. Notwithstanding, fishing (especially small-scale) plays a vital role in health nutrition, food security, and economic development of humans, especially coastal societies [9,10]. Fish meat provides nearly 20% of animal protein intake to up to one-third of the human population; with small-scale fisheries contributing two-thirds of this amount [9,11].

Despite the profound importance of fishing in society, fish are still viewed in market analyses as commodities rather than local food sources [7,9]. This means that the interdependencies between aquatic and terrestrial ecosystems in terms of ecological and human health, food security, economically and culturally viable livelihoods, and community well-being are still disregarded [9].

### 1.2. Vulnerability of Marine Fish to Exploitation

There are more species of fish than of all the other vertebrate groups combined. This high diversity translates into species having a wide assortment of biological and ecological characteristics [12]. The combination of these characteristics in a species will result in some being more vulnerable to exploitation than others [5]. In general terms, vulnerability is determined by the interaction between the exposure to some threatening (extrinsic) drivers and the intrinsic ability of populations to respond or adapt to such threats [4,13]. Life-history and ecological traits, which have evolved to guarantee persistence to biotic and abiotic variability, are regarded important to estimate intrinsic vulnerability in marine fishes [5,6,14,15].

In regard to life history, species with high vulnerability often have low intrinsic rates of population growth [16]. However, as estimates of this parameter are hard to obtain for marine species, scientist typically use a set of measurable traits as proxies of vulnerability [16]. For instance, a larger maximum body size, higher longevity, later age and larger length at maturity, slower body growth rate and lower natural mortality rate, are indicatives of species being more vulnerable to exploitation [5,6]. Additionally, some ecological traits and behaviors in marine fish are directly related to their susceptibility to exploitation. Fish aggregating behavior at fixed times and locations for purposes of feeding or spawning, complex reproductive strategies like sequential hermaphroditism (i.e., protogyny and protandry), internal fertilization and parental care, restricted geographic range size, rarity, and occupancy of certain habitats (e.g., seamounts, coral reefs) are examples of these traits and behaviors [14,15,17,18].

Understanding the patterns and process of vulnerability in marine fishes becomes necessary for fisheries management. The first species to be depleted are usually those with higher vulnerability to exploitation (e.g., large predatory fish), followed by lower trophic level species (e.g., herbivorous fish), which are often less vulnerable [16,17]. This pattern has led to a decline in the mean trophic level of global fish landings through time ('fishing down food webs'; [19]).

*1.3. Vulnerability, Extinction Risk and Threatened Species Lists in Marine Fish*

Estimates and correlates of vulnerability can be used to designate the conservation/fishery statuses of marine populations/stocks, when measured across time [4,20]. These, in turn, are used by scientists, fishery managers and conservation practitioners around the globe to, for instance, prioritize declining species for upcoming stock or extinction risk assessments, optimize resource allocation in species' recovery plans, inform reserve design, or report the state of the environment (e.g., assessment of the Convention on Biological Diversity's 2010 biodiversity targets; [21,22]). As a consequence, the continuous development of methods to estimate vulnerability can enhance our ability to effectively evaluate the status of marine fish populations and stocks, and guide decision-making in fisheries management and conservation [4,20]. Traditionally, vulnerability assessments have been used to create threatened species lists, in which declining species are categorized according to their extinction risk [20]. The International Union for Conservation of Nature and Natural Resources (IUCN) Red List is the most widely used globally [23], although there are others developed for local, national or regional levels, multiple political scales, or even specific taxonomic groups [2,21]. Category designation using IUCN criteria are based on parameters such as, a declining population from past or future projections, small population size, generation length, limited geographic range/distribution, extreme population fluctuations, or quantitative predictions of population viability. However, the use of life-history parameters as indicators of vulnerability is limited in these type of approaches [23–25].

In the past two decades, the use of biological traits (i.e., life history, ecology) to estimate vulnerability has gained popularity [26]. In the case of marine fish, one of the most popular approaches, developed by the American Fisheries Society (AFS), combines different life-history parameters to categorize distinct population segments (DPS) of a species according to their vulnerability to exploitation [27]. DPSs in marine fishes might encompass single stocks, groups of stocks, metapopulations, subspecies, or species, depending on available information [27]. The method ranks DPS according to their resilience (the opposite of vulnerability), in terms of intrinsic rates of population growth or other life-history traits (if the former cannot be estimated). Then, the method lists the DPS as vulnerable if their population declines as a consequence of fishing (or other threats), reaching a specific threshold assigned to the productivity category [4,16,27]. The AFS listing is mainly limited to US stocks; yet, the method to estimate resilience and assign categories was adapted to the FishBase database [28] under the name of 'Resilience' indicator [29].

*1.4. Vulnerability and Decision-Making in Multispecies Fisheries*

The ecological context of vulnerability may be important for decision making, but in fisheries management, the economic, political and social context matter too [2]. Even though stock assessments can help decision-makers to create strategies to achieve management objectives for each fishery, there is often insufficient time, resources and expertise to evaluate multispecies assemblages that occupy large areas in detail [2]. This is especially relevant in tropical multispecies fisheries, where a wide range of species are targeted or caught as by-catch, and fisheries value is low in global terms (though significant for local food security and livelihoods; [22–30]). Moreover, species caught by these fisheries are often data-poor in comparison to the ones in temperate areas or fisheries of global commercial value. Thus, the conservation/fishery statuses of these populations/stocks are often unknown [14,22].

In this regard, guiding species prioritization in tropical multispecies fisheries based on the application of common vulnerability assessments that estimate extinction risk (e.g., IUCN Red List, AFS method), might not be recommendable. This, as IUCN criteria can only be applied in species with sufficient information (otherwise they are categorized as 'Data Deficient'), and the scope of the assessment is global rather than local (although there are limited regional assessments) [2,4]. Additionally, thresholds of extinction risks usually conflict with the standard reference point criteria used in stock assessments [31]. By contrast, the AFS method can be applied to fish stocks; yet the criteria used to calculate resilience are hard to apply and can be ambiguous for data-poor species. This often leads to an underestimation of extinction risk in these species [4,15].

On the other side, there are methods that provide an estimate of vulnerability to fishing without necessarily assessing extinction risk. These methods make use of the 'precautionary principle' [32] as they categorize species according to the biological characteristics that make them susceptible to fishing pressure, rather than characteristics that are indicative of the path to extinction [24]. This way, we would be able to identify vulnerable species before they start to decline and implement suitable precautionary management strategies to prevent their extinction [24]. One example is a quantitative method developed by [15] (i.e. Cheung et al. 2005), which uses a 'fuzzy logic expert system' to combine all available information on key life-history and ecological traits (i.e., geographical range size, and strength of site fidelity during spawning or feeding), in order to generate a vulnerability score. This score is then used to assign species in a nominal vulnerability category. The assessment has also been incorporated into the FishBase database as a 'Vulnerability' indicator [29].

Other methods, like the Productivity Susceptibility Analysis (PSA), generate an index of species vulnerability, which is then assigned to a nominal risk category [33]. Risk in this case is not related to extinction, as in former assessments, but to exploitation. In this sense, this method does not only include biological correlates to estimate species (intrinsic) vulnerability, but also parameters that correlate to their susceptibility to a particular fishery (i.e., extrinsic vulnerability; [22,34,35]. Even though, some changes in productivity (resilience) parameters can occur as a response of fishing (e.g., reduction of age at maturity), these are less flexible than the susceptibility parameters. This way, opportunities to reduce risk by management measures, would be driven by changes in susceptibility, rather than productivity [22]. This method uses a semi-quantitative approach, as it assigns risk scores to each parameter depending of the established criteria (i.e., 1—low; 2—medium; 3—high), and then averages those values to obtain a vulnerability index [2].

PSA has been used in different applications for fisheries management. For instance, to prioritize management interventions in bycatch species of the Australian Northern Prawn Fishery [34], and to group data-poor stocks across USA waters into relevant management complexes [35]. A more interesting application was developed to evaluate different management scenarios in multispecies trawl fisheries of South East Asia [22]. Finally, this method has also been integrated in an ecological risk assessment framework, termed Ecological Risk Assessment for the Effects of Fishing (ERAEF; see [33,36] for examples). These examples illustrate that PSA is well-suited to evaluate ecosystem-based fisheries management (EBFM) objectives, when interactions and trade-offs between different ecological components of a fishery are important (i.e., target species, bycatch species, ecological communities, habitats; [7,37]).

*1.5. High Vulnerability in Grouper Species*

Groupers (Epinephelidae) are heavily exploited fish in tropical and subtropical coastal and benthic ecosystems [31]. According to the Food and Agriculture Organization (FAO), they contributed approximately 275,000 tons to global capture fisheries production in 2009, with an increment of 25% from the previous decade [31]. Larger groupers are targeted as luxury food, highly valued for taste or texture, and are taken for the live reef food fish trade [2]. Although some species are hatchery produced, this industry relies on juveniles taken from the wild [38]). Smaller species are also known to be impacted from by-catch in tropical multispecies fisheries [2]. In addition to their importance in global fisheries, they have a key role in marine ecosystems, principally in coral reefs. Groupers are predatory fish, and many species larger than 1 m in length are known to play an important role in moderating the abundance of prey species [31]. In addition, some species are considered ecosystem engineers (e.g., red grouper, *Epinephelus morio*), as they form burrows in barren soft substrates and create shelter for fishes and invertebrates [2].

Exploitation has been regarded as the main threat for most members of this family, particularly through direct mortality. It has been demonstrated that 84% of the global biomass of groupers has been removed from tropical (and temperate) reefs [39] by overfishing, and several species, like the Atlantic goliath, Nassau, Warsaw and potato groupers and the speckled hind (*Epinephelus itajara*,

*E. striatus*, *E. nigritus*, *E. tukula*, *E. drummondhayi*, respectively) have been severely depleted in parts of their range [2]. Additionally, coastal urban development and catastrophic natural events can also have a disproportionate effect on species with small range and population size (e.g., island grouper, *Mycteroperca fusca*; [31]). By 2013, 12% of species were considered as threatened, and 13% as near threatened (13%) according to IUCN criteria (out of 163 analyzed). Additionally, 30% were categorized as Data Deficient [31].

Most grouper species tend to be vulnerable to overfishing because of their biology; many have a large body size, are long-lived, reach maturity late, are sequential hermaphrodites, and spawn in aggregations that are targeted [38]. However, studies have concluded that body size is the best vulnerability predictor in groupers, as those species that decreased in abundance compared to their nearest relatives had a greater maximum size [14]. This relationship between body size and vulnerability is evident within most families, but strongest in groupers [16]. Additionally, members of this family take the longest time to increase population size and attain carrying capacity (between 20 to 40 years, depending on site-specific conditions), after the implementation of no-take reserves. Studies suggest that fishing, at even relatively low intensities (e.g., removal of 10% of standing stock), can push back recovery in groupers to more than a decade [16].

### 1.6. Aims of the Study

This review article employs the case study of the sailfin grouper (*Mycteroperca olfax*; hereinafter 'bacalao', as is locally known), an overexploited species targeted in the artisanal handline fishery of the Galapagos Islands [40], in order to illustrate the application of vulnerability assessments in marine fish in order to guide decision-making in multispecies fisheries. For such purpose, we first provided a context of the fishery, in which we examined (i) the historical perspectives and ecological impacts of the Galapagos' handline fishery, (ii) vulnerability, conservation status and drivers of overexploitation in bacalao and (iii) the progresses in monitoring and research in the fishery, and their contribution to fisheries management. We used PSA to estimate vulnerability and risk of exploitation in bacalao, and other phylogenetically-related species caught in the handline fishery, in order to prioritize species for management intervention. Using the same method, we evaluated hypothetical management scenarios aiming to recover bacalao stocks, by tracking changes in risk of exploitation in all the species after the implementation of these measures. Finally, we analyzed the viability of implementing such interventions by considering the socio-economic, political and governance aspects of the fishery. Results of this analysis will provide new insights on how to improve sustainability in bacalao and other grouper fisheries around the world.

## 2. Materials and Methods

### 2.1. Study Area

The Galapagos Islands are an archipelago located 1000 km approx. west of mainland Ecuador in the eastern Pacific Ocean (1°40′ N–1°36′ S, 89°16′–92°01′ W). They encompass thirteen islands (>10 km$^2$) and over 100 islets [41]. The Galapagos Islands are famous for their unique biodiversity, where nearly 20% of their marine species are endemic [42]. This happens as climatic and oceanographic conditions allow the co-existence of tropical, temperate and Southern Ocean species [43,44]. The need to preserve this unique environment encouraged its designation as Galapagos National Park in 1959, and a UNESCO World Heritage Site in 1979 [45].

In 1998, the Galapagos Marine Reserve (GMR) was created as a multiuse reserve under the newly issued Galapagos Special Law; an area of 138,000 km$^2$ that extends 40 nautical miles offshore from the baseline of the archipelago [45]. The GMR represents the major fisheries management tool in the present. In 2000, the zonation plan for the coast (shore to 2 nautical miles offshore) was agreed under its co-management system (which includes fisheries, tourism, research, NGOs and government representatives), although its limits were not demarcated until 2006. Within the limited-use zone of the

marine reserve, fishing is permitted in 78% of this coastal area, with the remaining 22% made up of several conservation and tourism areas (both no-take zones; [45–47]). In all open waters (>2 nautical miles from the shore), fishing is also allowed [48]. Currently, a new zonation scheme, which also includes the closure of offshore waters, was declared by the Ecuadorian legislature in 2016 and is in the process of being fully implemented [49].

*2.2. Species Prioritization in the Galapagos' Handline Fishery Using PSA*

Productivity Susceptibility Analysis (PSA) was employed to aid species prioritization for management intervention in the Galapagos' handline fishery. To accomplish this prioritization, we calculated vulnerability to fishing in bacalao and other species being caught within the same fishery. Vulnerability was used then, to assign species into categories of risk of exploitation, which could be employed for species prioritization. We focused on bacalao's phylogenetically-related species (i.e., groupers and serranids; hereinafter related species) in this analysis, as they share similar intrinsic vulnerability [31], and it was assumed that the handline fishery was causing a similar impact on these species. We used [50,51] to select related species, from which 7 were groupers, and 3 were serranids (Table 1).

To calculate vulnerability with PSA, we adapted the method applied in [33] (hereinafter reference study), as it was used for tropical multispecies fishers, including groupers. Calculations were conducted using the Risk-Based Framework worksheet in Excel available from the Marine Stewardship Council website [52]. Formulae to calculate the productivity index, susceptibility index, vulnerability (PSA score) and risk of exploitation, are available in Supplementary Materials (Table S1). In regard to productivity indices, we used the same life-history attributes (e.g., average size at maturity, maximum age, reproductive strategy) and risk score criteria as our reference study. For bacalao, we used estimates for these attributes available at [53], which represent long-term data of this fish stock in the Galapagos. For the related species, we mostly used estimates available in the FishBase Life-history tool [29], with the exception of some species having maximum lengths and fecundity (in the case of camotillo; *Paralabrax albomaculatus*) reported for the Galapagos (see [50,54]; Table 1). For the latter species, there was also an estimate of size at first maturity, though it represented the length at which 100% female fish reach sexual maturity, instead of 50% (which is normally considered as average size at first maturity). For such reason, we chose the one available in FishBase instead.

In the case of susceptibility indices, we also used the same attributes as the reference study but adapted the risk score criteria for the particular case of our fishery (Table 2). For availability (or areal overlap), in order to calculate the extent of the fishery, we assumed that the handline fishery targets all areas where bacalao is distributed around the archipelago. These encompass several seamounts south of Santa Cruz and coastal waters above 100 m of depth off the coastline of major islands, except the north of Isabela and southwest of Fernandina (see [55]). Following, we subtracted the area of the no-take zones included in the current GMR zonation scheme (tourism and conservation areas; [45]). We did this, as bacalao has traditionally been considered the main target species in this type of fishery (see below). We used [50] to estimate the relative species' distribution in the archipelago and then, we calculated the overlap of the fishery with their distributions. For instance, species living only in the northern islands had less overlap with the fishery than those with a broad distribution, as most coastal waters in the northern islands have been declared as no-take zones. For encounterability (or vertical overlap), we calculated the overlap of the gear depth range (15–200 m; [56]) over the species' depth range (taken from [25]). For selectivity, we accounted for the multiple hook sizes (up to 70 mm in length; [57]) used in handlines and assumed that the gear captures fish from 18 cm upwards (see [53]). Therefore, we inferred susceptibility to gear selectivity by using species' average size at maturity and maximum size, as both parameters are indicative of the effects that the gear would cause on fish populations (e.g., recruitment and/or grow overfishing; see [40]). For instance, species with a maximum size $\leq 18$ ($\pm 5$) cm had the lowest susceptibility to gear selectivity; whereas species with a size at maturity $\leq 18$ ($\pm 5$) cm were less susceptible than species with a larger one. Finally, for the post-capture

mortality parameter, we used similar criteria from our reference study. Bycatch or target species were less vulnerable when they are released as response of non-marketable size, as opposed to target species that are always retained regardless of their size (i.e., released species were identified as those having a bycatch ratio > 0, according to [56]). We found no evidence of post-capture survival after release in the literature; thus, no species was assigned to the lowest risk category.

### 2.3. Evaluation of Hypothetical Fisheries Management Scenarios Using PSA

PSA was then used to evaluate hypothetical fisheries management scenarios aiming to recover bacalao stock, and to analyze the implications for the related species. As handlines are generally considered a non-selective gear (see Section 3.1), the implementation of regulatory management measures that, for instance, modify gear selectivity (e.g., alter hook size limitations), or restrict fishing in certain times or areas (e.g., seasonal closures, rezoning of fishing areas) would eminently have consequences on other species caught by the fishery (as either target or bycatch species). Thus, the scenarios could be used as a base to implement precautionary management measures for some of these related species that are highly vulnerable to fishing but have insufficient data to assess population declines.

The management scenarios that we evaluated include: (S2) enforcing a minimum (≥65 cm TL) and maximum (≤78 cm TL) landing size for bacalao (i.e., suggested in [40]); (S3) imposing larger hook sizes for handlines, so the minimum size caught is around 65 cm (i.e., age of first maturity in bacalao; Table 1); (S4) including additional no-take zones in coastal waters of the GMR [49], that comprise key nursery habitats for bacalao (e.g., mangroves, sandy coastal lagoons, rocky reefs, lava pools; [55]) and (S5) S4 plus the closure of spawning aggregation sites for bacalao (S5). We also included combined scenarios: (S6) sum of S2 and S3; (S7) sum of S2 and S4 and (S8) sum of S3 and S4. For each management scenario, we modified the susceptibility risk scores in bacalao and related species, and recalculated vulnerability indices (see Supplementary Materials; Table S2). When evaluating these management scenarios, we tracked displacements in the categories of risk of exploitation from the state condition (S1; i.e., first scenario).

**Table 1.** Summary of groupers and serranids commonly caught in the Galapagos' handline fishery (as target or bycatch species) and their characteristics. The latter include IUCN category (with date of last assessment), FishBase Resilience and Vulnerability categories, productivity attributes (4th–10th column), distribution within the islands, fishery importance and usual depth range. Within fishery importance, Retained means that fish are usually kept by fishers, whereas Released means that fish are usually returned to the sea as response of unmarketable size or another reason. BS = broadcast spawner; YLA = year of last assessment. If a superscript is placed in a column heading, all entries are taken from that reference, except when indicated.

| Scientific Name | English Name | Age at First Maturity (year) [a] | Maximum Age (year) [a] | Fecundity (egg/ind/year) | Maximum Length (cm) | Size at Maturity (cm) [a] | Reproductive Strategy [a] | Trophic Level [a] | Fishery Importance [d,e] | Distribution Galapagos [d] | Depth Range (m) [f] | IUCN Red List Category & YLA [f] | FishBase Resilience Category [a,g] | FishBase Vulnerability Score & Category [a,h] |
|---|---|---|---|---|---|---|---|---|---|---|---|---|---|---|
| *Mycteroperca olfax* | Sailfin grouper | 6.5 [b] | 21 [b] | - | 100 [b] | 65.3 [b] | BS | 4.2 [b] | Target Retained | Entire archipelago | 5–100 | VU (2016) | Medium | 56 (High) |
| *Alphestes immaculatus* | Pacific mutton hamlet | 2.1 | 8.1 | 10,000 [a] | 30 [a] | 18.5 | BS | 3.5 | Bycatch Released | Entire archipelago | 3–32 | LC (2016) | High | 32 (Low to moderate) |
| *Cephalopholis panamensis* | Pacific graysby | 3.6 | 14.2 | - | 39 [a] | 17.7 | BS | 4.2 | Bycatch Released | Entire archipelago | 1–30 | LC (2016) | Medium | 39 (Moderate) |
| *Dermatolepis dermatolepis* | Leather bass | 5.6 | 26.2 | - | 77 [d] | 53.5 | BS | 4.5 | Target Released | Entire archipelago | 5–50 | LC (2017) | Very low | 64 (High) |
| *Epinephelus cifuentesi* | Olive grouper | 2 | 9.6 | - | 75 [d] | 62 | BS | 4 | Target Released | Northern islands | 40–120 | LC (2016)VU (2004) * | Medium | 39 (Moderate) |
| *Epinephelus labriformis* | Starry grouper | 3 | 11.8 | - | 60 [d] | 21.1 | BS | 4 | Bycatch Released | Entire archipelago | 1–50 | LC (2016) | Low | 40 (Moderate) |
| *Hyporthodus mystacinus* | Misty grouper | 8.2 | 41.3 | - | 136 [d] | 81.1 | BS | 4.6 | Target Retained | Northern islands | 12–400 | LC (2016) | Very low | 85 (Very high) |
| *Paranthias colonus* | Pacific creole-fish | 2.4 | 9.5 | - | 35.6 [a] | 21.5 | BS | 3.8 | Bycatch Released | Entire archipelago | 0–120 | LC (2016) | Medium | 34 (Low to moderate) |
| *Cratinus agassizii* | Graery threadfin seabass | 3.6 | 15.9 | - | 61 [d] | 34.1 | BS | 4.2 | Target Retained | Entire archipelago | 1–25 | NT (2017) | Medium | 49 (Moderate to high) |
| *Hemilutjanus macrophthalmos* | Grape-eye seabass | 3.7 | 15.9 | - | 50 [a] | 29 | BS | 4.3 | Bycatch Released | Entire archipelago | 10–55 | DD (2017) | Medium | 53 (Moderate to high) |
| *Paralabrax albomaculatus* | White-spotted sandbass | 3.7 | 15.9 | 542,000 [c] | 64 [c] | 29.4 | BS | 4.5 | Target Released | Western Isabela & Fernandina | 10–75 | EN (2007) | Medium | 53 (Moderate to high) |

a [29]; b [53]; c [54]; d [50]; e [56]; f [25]; g [27,28]; h [15]; * IUCN Regional Assessment for the Galapagos.

**Table 2.** Cutoff criteria used for assigning scores in susceptibility parameters for a Productivity–Susceptibility Analysis in the Galapagos' handline fishery. Score criteria were adapted from [33].

| Parameter | Low Risk (Score = 1) | Medium Risk (Score = 2) | High Risk (Score = 3) |
|---|---|---|---|
| Availability (areal overlap) | <10% overlap | 10%–30% overlap | >30% overlap |
| Encounterability (vertical overlap) | <45% overlap | 45%–85% overlap | >85% overlap |
| Selectivity | Maximum size ≤ 18 (±5) | Size at first maturity ≤ 18 (±5) | Size at first maturity ≥ 18 (±5) |
| Post-capture mortality | Released always and evidence of survival | Released when size is non-marketable | Retained always |

The color scale was applied to indicate level of risk.

## 3. Results and Discussion

### 3.1. Context of the Fishery: A Summary

#### 3.1.1. Historical Perspectives of the Galapagos' Handline Fishery and Ecological Impacts

Human exploitation of sea resources in the Galapagos started in the late 18th century by whalers and fur seal hunters. Currently, the fishing sector is second in importance in the local economy after tourism [58]. With the creation of the GMR, large-scale industrial fishing (e.g., pelagic longline) was banned, including capture of iconic species like sharks, whereas artisanal and subsistence fishing were re-organized by zones. The main artisanal fisheries are sea cucumber (currently closed), lobsters, and finfish, which are carried out by simple, hand-operated gears (e.g., line and hooks or small nets), diving, or manual collection from the shore [59,60].

Finfish fisheries in the Galapagos ('pesca blanca') date back to the time of European colonization, when around twelve species were caught for subsistence [60]. Traditionally, the most utilized gear over rocky reefs has been the handline (locally known as 'empate'), which captures around 68 species of demersal fish [61]. This gear consists of a monofilament line weighted with lead and several short extensions of propylene line, each with one hook [62]. Handlines are used often between 15–200 m, from coastal waters up to open waters over seamounts [51,56]. The most iconic species captured by this gear are bacalao, misty grouper (*Hyporthodus mystacinus*), camotillo (*Paralabrax albomaculatus*) and mottled scorpionfish (*Pontinus clemensi*); yet, groupers usually dominate the landing composition [62,63]. Until the 1960s, bacalao and other groupers were mainly caught during the warm/wet season (December to April), and then salt-dried and exported to the mainland for the preparation of 'fanesca', a traditional Ecuadorian dish served at Easter. Nowadays, the fishery is open all year round to supply the local population and tourism sector with fresh fish, and mainland demand with fresh, frozen and salt-dried fish [56,62]. Currently, fishers can either perform single day trips in smaller vessels (pangas or fibras) and travel to nearby locations; or perform multiday trips with larger vessels ('mother') with one or more small vessels attached and travel to locations farther from the fishing ports. This has resulted in differences in landings across the archipelago: landings from small vessels mainly originate from the central zone, whereas landings from 'mother' vessels usually originate from the western and northern zones [40,62].

Handline gear is one of the most selective fishing gears used by small-scale fishers, being fairly selective towards large apex predators [64,65]. Larger-sized hooks often increase catch rates and narrow down selection ranges in comparison to smaller-sized hooks [65]. However, in the Galapagos, where a maximum length in hooks is imposed (70 mm; [57]), a high proportion of the catch (e.g., 0.40 in terms of biomass; [56]) is usually composed of noncommercial species, nonmarketable sizes or regulated species, like sharks. Therefore, handline has been regarded as a low-selectivity gear in the islands [56,62]. Moreover, the frequent capture of large fish predators when employing this gear, can cause cascading effects in the marine ecosystem [64]. For instance, after El Niño 1982/1983, the removal of lobsters and fish predators caused changes in the composition and structure of fish assemblages, and the spread of grazing sea urchin populations, mainly *Eucidaris thouarsii* [66,67].

### 3.1.2. Bacalao: Vulnerability, Conservation Status and Drivers of Overexploitation

The sailfin grouper (*M. olfax* Jenyns, 1840) or bacalao is endemic to the Eastern Tropical Pacific and has a restricted geographic range that includes the Galapagos Islands, Cocos Island (Costa Rica) and Malpelo Island (Colombia) [68,69]. As other members of the grouper family, it has a large maximum body size, long life-span, slow growth rate and delayed sexual maturity [31,40,53] (Table 1). Additionally, the fish appears to be protogynous, exhibiting a highly female-skewed sex ratio (0.009 male per female); though this pattern might be the result of complex social behaviors, in which males and females are segregated during the non-reproductive season [53,70]. Reproduction can occur throughout the year, but peaks have been identified between October and January [53,71]. In addition, during the reproductive season the species forms spawning aggregations, as other members of the family [71]. Finally, trophic level in the species was estimated as 4.2, indicating its role as an apex predator in rocky bottom ecosystems of the Galapagos [72]. Ecosystem models have predicted that the loss of this species could trigger cascade effects onto lower trophic levels [64,73].

Previous [74] and current [40] stock assessments have demonstrated clear signs of growth and recruitment overexploitation in bacalao. One assessment conducted by [75] concluded that this species is not overexploited, but many of his conclusions are considered incorrect in the present [76]. Based on landings data and levels of exploitation, it is inferred that the population has declined by at least 30% over the past 40.5 years. In response, bacalao was listed as 'Vulnerable' during the last IUCN assessment (Table 1) and has maintained this category since 1996 [69].

It is important to mention that the socio-economic drivers of exploitation for the species have changed through time. For decades bacalao was the most valuable species in the handline fishery, comprising almost the totality of the finfish landings around the 1940s [74]. Fishers often targeted spawning aggregations, allowing them to catch a large proportion of the reproductive population of the species in a brief period of time. During that time, all production was salt-dried, and then exported to the mainland, in order to supply the high demand of fish for Easter [77,78]. After the prominent human population increase in the islands in the 1970s, fishers started to catch bacalao all year round, but also other demersal fish, in order to supply the tourism industry and local demand. As a result, by the end of this decade, bacalao only represented 36% of the total finfish catch. This reduction was probably caused by population decline, but also due to other species becoming more commercial in the local market (e.g., misty grouper *Hyporthodus mystacinus*) and the exports of tuna. After 2010, this percentage was further reduced to <20% [62,74,77].

Despite the signs of overexploitation in the species, including a reduction in fish size over time [40], catch rates have remained fairly stable in the last decades. This is likely a consequence of the fishery expanding to other parts of the islands and the rise of tracking technologies for fish (e.g., probes, positioning systems, larger engines; [51,61]. Moreover, the fact that new generations of fishers do not perceive these declines (i.e., 'shifting baselines syndrome'; [79]), prevents economic extinction, as fishers keep targeting the species above sustainable levels [80].

Weaknesses in political and governance aspects in the artisanal fisheries of the Galapagos, have also prevented the bacalao stock to recover over time [40,61]. One of the main issues involves the lack of credibility and legitimacy in the co-management system of the GMR [81,82]. The Participatory Management Advisory Council (previously called Participatory Management Board) encouraged the involvement of fishers in the 2000 zoning scheme of the marine reserve, and the creation of instruments, like the fisheries management plan (known as 'Capítulo Pesca' in past years, currently 'Calendario Pesquero Quinquenal'; [59,60,83]); though, it failed in accomplishing this [84]). In addition, these instruments do not include any specific management measure for bacalao (e.g., maximum landing size, seasonal closures), other than zonation, a licensing system, custody chain, and list of allowed fishing gears [59,60]. Another issue is the lack of compliance among fishers, who frequently incur into illegal activities, like using spearguns, or fishing inside no-take zones [61]. The existence of overlapping boundaries among different subzones of the GMR has aggravated the latter situation [48,84]. In addition, control and enforcement is difficult in the GMR, as surveillance is costly and challenging,

and sanction rates and penalties are undervalued [40,85,86]. One thing to consider is that by the time these management instruments and the 2000 zonation system were implemented, bacalao already showed signs of overexploitation, with scarce signs of recovery [84,85].

### 3.1.3. Progresses of Monitoring and Research in Bacalao Fishery, and Their Contribution to Fisheries Management

Even though bacalao fishery started around the 1920s, fisheries' monitoring and research did not start until the 1960s [40,77]. In fact, the development of marine and fishery science in Galapagos was not considered an immediate priority until two decades ago [81]. Fisheries-dependent population data for bacalao (e.g., biomass composition in landings, catch per unit effort, total catch volume, landing length) are only available from the periods of 1977–81 [74], 1988–90 [87], 1997–2003 [62], 2006 [51], 2009 [75], 2011–2013 [53,56], although data vary in methodology and spatial coverage. Monitoring during the period 1997–2003 was carried out by the Participatory Fisheries Monitoring and Research Program (or PIMPP in Spanish), and involved fishers' participation [81]. These data allowed scientists to conduct stock assessments for the species at different periods (see [40,74,75]).

Fishery-independent population data is also patchy and fairly recent. Studies like [64,75,88–90], provide estimates of population density (and sometimes biomass and total length) for bacalao in the adult stage across bioregions and across management types (i.e., take and no-take zones; see Table 3 for density comparisons). Other studies have found a large dependence of juvenile bacalaos to mangrove ecosystems [91,92]. Since 2000, the Ecosystem Research and Monitoring Program (PIMEC in Spanish) conducted by the Charles Darwin Foundation and Galapagos National Park Service implemented a standard method to monitor subtidal rocky bottoms using standard underwater visual methods [93]. Some studies have published results using this method (e.g., [88–90]), and have been used, in part, to create a habitat suitability map of the species in the Galapagos [55]. All these data and studies show that bacalao is widespread across the entire archipelago, but its density varies among different bioregions. The western bioregion holds highest densities of this species, followed by the central-southeastern (Table 3). Unfortunately, after more than 20 years of marine reserve implementation, scientists have not found higher densities (and biomass) of the species in no-take vs. take zones [69] (Table 3). However, Ref. [94] demonstrated that at locations where fishing is prohibited in the GMR, there is a higher biomass of apex predators (including bacalao).

**Table 3.** Density estimates (standardized to No. individuals/100 m$^2$) for adult bacalao (*Mycteroperca olfax*) in different studies, from 1991 to 2014, using underwater visual censuses. Comparisons were carried out across management zones (open vs. closed fishing areas) and bioregions [89]. From 2000, management zones represent the implemented zoning plan of the Galapagos Marine Reserve; whereas in previous years, they represent averages in lightly fished vs. highly fished areas. Values with double superscripts represent averages of the density estimates from two datasets.

| Year | Management Zones | | Bioregions [89] | | | | |
| | Open/Highly Fished | Closed/Lightly Fished | Far-Northern | Northern | Central-Southeastern | Western | Elizabeth |
|---|---|---|---|---|---|---|---|
| 1991 | 0.17 [1] | 0.13 [1] | | 0.33 [1] | 0.12 [1] | 0.13 [1] | 0.10 [1] |
| 1998 | 0.04 [2] | 0.51 [2] | | | 0.28 [2] | | |
| 2000 | 0.52 [4] | 0.58 [4] | 0.21 [3,5] | 0.84 [3,5] | 2.18 [3,5] | 1.65 [3,5] | 0.86 [3] |
| 2001 | 0.52 [4] | 0.58 [4] | 0.26 [3,5] | 1.14 [3,5] | 1.49 [3,5] | 2.92 [3,5] | 0.86 [3] |
| 2002 | | | 0.36 [5] | 0.36 [5] | 1.01 [5] | 1.72 [5] | |
| 2003 | | | 0.58 [5] | | 0.88 [5] | 4.27 [5] | |
| 2005 | | | 0.55 [5] | 0.61 [5] | 0.70 [5] | 2.18 [5] | |
| 2006 | | | | 0.48 [5] | 0.55 [5] | 2.27 [5] | |
| 2007 | | | 0.40 [5] | 0.68 [5] | 1.35 [5] | 3.31 [5] | |
| 2008 | | | 0.64 [5] | 0.48 [5] | 0.98 [5] | 4.79 [5] | |

[1] [88]; [2] [64] [3] [89]; [4] [90]; [5] [75].

On the other hand, there are still several gaps of information about ecology and life history in bacalao, including larval biology and dispersal, connectivity between ontogenic habitats, population genetic structure, spawning aggregation sites, natural predators, natural mortality rates, and the influence of physical factors in population dynamics [55]. Additionally, there has not been enough focus in understanding socio-economic aspects of the fishery, such as commercialization channels, optimal fishing effort, linkages between fishing and tourism, the role of women in the fishery, among others [55,81]. Finally, there is a need to study the impact of other threats on the species (other than fishing), such as climate change and climate variability, habitat degradation and pollution (especially in juvenile habitat), and marine invasive species.

Despite the knowledge gaps in the fishery, scientists have concluded that there is enough information to estimate local vulnerability, and to implement suitable management strategies that could promote stock recovery [76]. In such regards, a recent study conducted by [53] updated estimates of important life-history parameters for bacalao (e.g., growth rate, age and length at fist maturity, longevity, sex ratio), and suggested that the species is even more vulnerable to exploitation than we originally thought. These parameters were afterwards used by [40] to conduct the latest stock assessment of the species, and to recommend bacalao-specific management regulations that need to be implemented in conjunction with other measures (i.e., rezoning of the GMR to protect essential bacalao habitat, including key nursery habitats and spawning aggregations sites) to guarantee stock recovery. These specific management regulations comprised technical measures as minimum landing size ≥ length at maturity (65 cm TL), maximum landing size ≤ to mean size of mega-spawners (78 cm TL), or slot limits (~64–78 cm TL) and effort control measures like seasonal closure during peak spawning (October to January) [40].

Although a combination of these regulatory management measures is essential to revert negative population trends in the species, this does not mean that there are not already efforts being tested or implemented for bacalao. These measures include for instance, the creation of agreements between fishers and tour operators to shorten supply chains, or to sell processed products (e.g., smoked fish) [85]; the implementation of improved origin certification and traceability schemes [95] and the deployment of fishing aggregation devices to attract fast-growing pelagic fish (like tuna and wahoo) and reduce fishing pressure on bacalao [95]. However, there is no available information about the outcomes of these interventions.

*3.2. Vulnerability and the Implications for Decision-Making in the Bacalao Fishery*

3.2.1. Improving Species Prioritization in Groupers and Serranids Caught by the Galapagos' Handline Fishery

The PSA showed that bacalao was the most vulnerable species among selected groupers and serranids caught in the Galapagos' handline fishery (score of 3.61), and the only one assessed as having a high risk of exploitation under the current state of the fishery (Table 4). This is not surprising, as the species has already been signaled as vulnerable in FishBase (e.g., 'High' vulnerability, with a score of 56; Table 1). In our analysis the species had the second lowest productivity (index of two), but it was the most susceptible species to exploitation (index of three). Other species like the misty grouper (score of 2.87), leather bass (*Dermatolepis dermatolepis*; score of 2.74) and camotillo (score of 2.73) were assessed as having a medium risk of exploitation, while the remaining species were assessed as having a low risk of exploitation (Table 4). The misty grouper and the leather bass were assessed as vulnerable in our analysis in response of their low productivity (indices of 2.17 and two, respectively). The camotillo, instead, is a species of medium productivity (1.43), but it is more susceptible to the fishery (index of 2.33) than the groupers. In fact, it is a globally endangered species according to the IUCN Red List, as a result of a significant population decline and very restricted geographical range (endemic to the Galapagos).

**Table 4.** Productivity–Susceptibility Analysis results (PSA) (vulnerability index and risk of exploitation) for bacalao (*Mycteroperca olfax*) and other groupers and serranids commonly caught in the Galapagos' handline fishery. Parameters of productivity and susceptibility were taken and adapted from [33].

| Scientific Name | Age at maturity | Maximum Age | Fecundity | Maximum Size | Size at Maturity | Reproductive Strategy | Trophic Level | Productivity Index | Availability | Encounterability | Selectivity | Post-Capture Mortality | Susceptibility Index | PSA Score (Vulnerability) | Risk of Exploitation |
|---|---|---|---|---|---|---|---|---|---|---|---|---|---|---|---|
| *M. olfax* | 2 | 2 | - | 2 | 2 | 1 | 3 | 2.00 | 3 | 3 | 3 | 3 | 3.00 | 3.61 | High |
| *A. immaculatus* | 1 | 1 | 2 | 1 | 1 | 1 | 3 | 1.43 | 3 | 2 | 2 | 2 | 1.58 | 2.13 | Low |
| *C. panamensis* | 1 | 2 | - | 1 | 1 | 1 | 3 | 1.50 | 3 | 2 | 2 | 2 | 1.58 | 2.18 | Low |
| *D. dermatolepis* | 2 | 3 | - | 1 | 2 | 1 | 3 | 2.00 | 3 | 2 | 3 | 2 | 1.88 | 2.74 | Med |
| *E. cifuentesi* | 1 | 1 | - | 1 | 2 | 1 | 3 | 1.50 | 2 | 3 | 3 | 2 | 1.88 | 2.40 | Low |
| *E. labriformis* | 1 | 2 | - | 1 | 1 | 1 | 3 | 1.50 | 3 | 2 | 2 | 2 | 1.58 | 2.18 | Low |
| *H. mystacinus* | 2 | 3 | - | 2 | 2 | 1 | 3 | 2.17 | 2 | 2 | 3 | 3 | 1.88 | 2.87 | Med |
| *P. colonus* | 1 | 1 | - | 1 | 1 | 1 | 3 | 1.33 | 3 | 3 | 2 | 2 | 1.88 | 2.30 | Low |
| *C. agassizii* | 1 | 2 | - | 1 | 2 | 1 | 3 | 1.67 | 3 | 1 | 3 | 3 | 1.65 | 2.35 | Low |
| *H. macrophthalmos* | 1 | 2 | - | 1 | 1 | 1 | 3 | 1.50 | 3 | 2 | 3 | 2 | 1.88 | 2.40 | Low |
| *P. albomaculatus* | 1 | 2 | 1 | 1 | 1 | 1 | 3 | 1.43 | 3 | 3 | 3 | 2 | 2.33 | 2.73 | Med |

The color scale was applied to indicate level of risk.

Nonetheless, it is important to take into account that our PSA vulnerability scores are only relevant for the context of this particular fishery, and that most species are also caught using other legal and illegal gears in the Galapagos (e.g., superficial hook and lines, Hawaiian slings, lures, spearguns). This means that species prioritization might vary depending on the type of fishery that we choose. Moreover, it was expected that our results would be different from other vulnerability assessments which categorize species mainly according to their extinction risk (e.g., IUCN Red list); yet more similar the ones based on fish life-history and ecological traits (e.g., FishBase Resilience and Vulnerability indicators). However, we also found some discrepancies between our results and the latter. For instance, bacalao was categorized as a species of 'Medium' resilience in FishBase, despite having a low productivity in our analysis (Tables 1 and 4). A similar case occurs with the starry grouper (*Epinephelus labriformis*), which was considered as a species of 'Low' resilience in FishBase, but of medium productivity (index of 1.50) in our analysis (Tables 1 and 4). These differences might be the result of FishBase employing lower age at first maturity ($t_m = 3$) and maximum age ($t_{max} = 7$) values than our study for bacalao, and a higher a maximum age ($t_{max} = 23$) for the starry grouper [29].

Finally, despite the disparities of results among different vulnerability assessments, our results demonstrate that PSA can provide valuable information to support decision-making in areas with limited research and monitoring capacity in fisheries, like the Galapagos. As other indicators have been already used to evaluate fisheries in the archipelago (see [83] for spiny lobster fishery's indicators), those could be included as parameters/attributes in PSA, and replace the ones typically used. Likewise, larger decision-support tools in fisheries and environmental management (e.g., Drivers-Pressure-State-Impact-Responses framework) could incorporate parameters and results of PSA, as was the case of the ERAEF [33,36]. Finally, the method itself, could be used to identify and fill information gaps in a particular species. In our case, local life-history and ecological parameters were only used for bacalao (and in a lesser degree in camotillo); yet in other species, data were filled with the best available information (e.g., FishBase Life-history tool). Though, as new data arise in the Galapagos, estimates of vulnerability on the species could be updated and improved periodically.

3.2.2. Evaluation of Hypothetical Management Scenarios to Recover Bacalao Stock and the Implications for Related Species

PSA was additionally applied to evaluate hypothetical management scenarios that would support bacalao stock recovery. Our results suggest that in order to shift bacalao from high to low risk of exploitation, a major rezoning of the GMR would be necessary. This involves adding no-take zones in coastal waters and protecting spawning aggregation sites of the species (S5; Table 5). Implementing any of the other scenarios, would produce a less optimal outcome (e.g., bacalao shifts only from high to medium risk of exploitation). At the same time, if our objective is recovering bacalao stock while creating a positive impact on other species, the most feasible scenarios would still be the same (S5), as species like the leather bass, misty grouper and camotillo would shift from medium to low risk of exploitation. This was accomplished with this scenario, as all species were given a lower availability risk score, as a result of a lower overlap of the fishery with their distribution. Creating additional no-take zones only in coastal waters (S4), or implementing combined scenarios like S7 (i.e., minimum and maximum landing size and S4) or S8 (sum of larger hook size for handlines and S4), would produce the same benefit for bacalao's related species; though, this grouper would only decrease its risk of exploitation from high to medium. This outcome is a result of a reduced areal overlap with the fishery (availability parameter; S4) and/or gear selectivity (Selectivity parameter; S3). However, as spawning aggregation sites for bacalao would still be unprotected, the fishery would still be able to catch a large proportion of the population. On the other hand, imposing a larger hook size for handlines (S3) or combining this scenario with minimum and maximum landing sizes for bacalao (S6), would only benefit the latter plus the leather bass and camotillo. The misty grouper would remain a medium-risk species, as it has the largest maximum size out of all the species considered; thus, a larger hook size would not affect gear selectivity towards this species. Lastly, setting minimum and maximum landing sizes (S2) would only benefit bacalao, but not other species, as this scenario would only reduce post-capture mortality for this species (Table 5). As noticed, any of our scenarios modified the encounterability parameter, as fishers would still be able to capture bacalao and other species in take zones of the GMR, across all vertical levels.

**Table 5.** Summary of results of Productivity–Susceptibility Analysis used to evaluate hypothetical management scenarios aiming to recover bacalao (*Mycteroperca olfax*) stock in the Galapagos Islands. Results are presented as risk of exploitation categories (high, medium, low) for bacalao and other related species (groupers and serranids) commonly caught in the Galapagos' handline fishery. GMR: Galapagos Marine Reserve.

| Management Scenarios | Description of Scenarios | Species | | | | | | | | | | |
|---|---|---|---|---|---|---|---|---|---|---|---|---|
| | | *M. olfax* | *A. immaculatus* | *C. panamensis* | *D. dermatolepis* | *E. cifuentesi* | *E. labriformis* | *H. mystacinus* | *P. colonus* | *C. agassizii* | *H. macrophthalmos* | *P. albomaculatus* |
| S1 | State condition | High | Low | Low | Med | Low | Low | Med | Low | Low | Low | Med |
| S2 | Set minimum (≥65 cm TL) and maximum (≤78 cm TL) landing sizes for bacalao | Med | Low | Low | Med | Low | Low | Med | Low | Low | Low | Med |
| S3 | Impose larger hook sizes for handline gear, so the minimum size caught is ~65 cm (age of first maturity for bacalao) | Med | Low | Low | Low | Low | Low | Med | Low | Low | Low | Low |
| S4 | Rezoning GMR: additional no-take zones in coastal waters, including key nursery habitats for bacalao | Med | Low | Low | Low | Low | Low | Low | Low | Low | Low | Low |
| S5 | Rezoning GMR: S4 + spawning aggregation sites for bacalao | Low | Low | Low | Low | Low | Low | Low | Low | Low | Low | Low |
| S6 | S2 + S3 | Med | Low | Low | Low | Low | Low | Med | Low | Low | Low | Low |
| S7 | S2 + S4 | Med | Low | Low | Low | Low | Low | Low | Low | Low | Low | Low |
| S8 | S3 + S4 | Med | Low | Low | Low | Low | Low | Low | Low | Low | Low | Low |

The color scale was applied to indicate level of risk.

Nevertheless, solely considering ecological viability when evaluating different management scenarios in fisheries could lead us to an inadequate answer, if we do not also contemplate the socio-economic and political (and governance) implications of implementing these interventions. Although our aim was not to provide an exhaustive and comprehensive evaluation of all possible management measures (regulatory and non-regulatory) to reduce vulnerability in bacalao, the successes and failures of different management measures in other groupers could give us an insight of which scenarios would be more suitable for our species.

Spatial closures (e.g., no-take zones in marine reserves) are considered the best management measure for sedentary fish species like groupers, especially when established by community efforts or they are included in the decision-making process [31]. For instance, groupers of the genus *Mycteroperca* have shown a 400% increase in biomass after 10 years of marine reserve implementation in Cabo Pulmo National Park, Mexico [96]. However, in the case of bacalao, lack of compliance in fishers has hindered species recovery in no-take zones (see Sections 3.1.2 and 3.1.3; Table 3). In response, since 2014, the Galapagos National Park Service has initiated the rezoning plan process for the GMR [48,86] providing an opportunity to improve bacalao recovery. According to the new zoning proposal (launched in 2016), a new marine sanctuary around Darwin and Wolf Islands will be created, along with 21 smaller no-take zones [49], which include key coastal habitats, like mangroves. The protection of these areas could revert the 'fishing down food webs' trend in locations where bacalaos are currently depleted [94]. Unfortunately, these areas are very sensitive to natural and anthropogenic pressures (e.g., pollution, climate change, ENSO events, among others; [97]); thus, mitigation measures should be included in new management plans of the marine reserve. In addition, managers should contemplate that the newly-proposed no-take zones could cause major exclusions to traditional fishing grounds to fishers; thus, provoke unrest among fishers.

The establishment of new no-take zones in open waters that host spawning aggregation behavior in bacalao, would increase reproductive potential, and decrease the loss of mega-spawners in the population. However, the biggest disadvantage is that most of these sites have not been identified yet (with the exception of few sites in the far-northern bioregion) [71], and they would require high costs and logistics for control and surveillance, as opposed to the ones in coastal waters. This underlines the urgent need for research on spawning aggregations of this species (see Section 3.1.3), as well as for other vulnerable groupers caught in the fishery. It has been observed, that the protection of spawning areas has prompted the recovery of local populations of species, like the red hind (*Epinephelus guttatus*) in the US Virgin Islands and Bermuda, the Nassau grouper in the Cayman Islands, and the camouflage grouper (*E. polyphekadion*) and squaretail coralgrouper (*Plectropomus areolatus*) in Palau [31].

In addition to spatial management measures, we believe that technical instruments, like minimum and maximum landing size are necessary to decelerate recruitment overfishing (i.e., harvesting too many fish before they have matured [98]), and to protect the large fecund males (mega-spawners) in the population, respectively [40]. Size limits have produced positive results, especially in large groupers with a similar protogynous reproductive mode as our species [31]. Nevertheless, even if this measure would stimulate post-capture release of individuals caught with handlines and other gear, it would not prevent post-capture survival. This, as fishers have reported that most individuals (across all size range) come aboard already dead as a response of barotrauma [99]. This means that in order to guarantee the effectiveness of this measure, it would be necessary to implement it in conjunction with others directed to increase the selectivity of the optimal sizes (e.g., changes in hook size, bait, sites, seasons, or a mix). In addition, stricter control in ports would be necessary to prevent the sale of fish outside of size limits.

In such regard, the imposition of larger hook sizes for the handline gear, could be effective to reduce recruitment overfishing in bacalao, as well as fishing pressure in small bycatch species, as they modify gear selectivity towards bigger and older fish (Omtomwa et al. 2019). This measure would probably result in a benefit to fishers' incomes, as bigger fish generate more revenue, and smaller fish species could still be caught using other gear. However, larger hooks have been proved to have

higher catch rates compared to the smaller ones [65]; thus, they could lead to an increased pressure towards large male fish in the population [40], and the incidental catch of large predators, such as sharks, sea lions and dolphins. In addition, it could encourage fishers to shift to other gears equally or more detrimental than handlines (e.g., trawl net, Hawaiian sling) to compensate for the losses.

After analyzing the socio-economic and political aspects of fisheries in the Galapagos, we believe that the most suitable scenario in the long term would be the inclusion of additional coastal waters and spawning aggregation sites for bacalao as no-take zones (S5). However, in order to accomplish a more thorough protection of essential habitat for this species, it would be necessary to increase knowledge about spawning aggregations (i.e., locations, seasonality), by using fishers' local ecological knowledge (LEK) or SCUBA surveys in the most important fishing grounds and seasons (assuming they encompass spawning areas). In such regard, it is likely that the most suitable scenario in the short-term would be the inclusion of additional coastal waters in the new zoning scheme of the GMR (S4), even though it does not result in the lowest reduction of risk of exploitation on the species. This, as some key nursey ground have already been identified on the species, especially in mangroves (see [91,92]), and the new zoning scheme would comprise some of these areas. Though, in order to estimate the potential effect on bacalao stock recovery, it would be necessary to calculate the area percentage of the new no-take zones that encompass these habitats.

At the same time, we are conscious that other regulatory management measures exist (e.g., total allowable catch, bag limits, effort quotas, individual transferable quotas) that could be suitable for bacalao. For instance, total closures (moratoria) are highly advisable when the exploited resources have limited resilience capacity, like most groupers [31]. Moratoria have prevented, for instance, the local extinction of the southeastern U.S. stock of the Atlantic goliath grouper [100]. However, we understand that this measure may not be realistic for bacalao, as the species holds a traditional value for people in the Galapagos [62,74], and past experiences with moratoria (e.g., sea cucumbers) have led to unrest among fishers [85]. On the other side, seasonal closure could prevent fishers from catching bacalao during peak spawning activity (October to January usually), but it could lead to a 'race for the fish' in other seasons or cause a shift of interest to other vulnerable species (e.g., leather bass, misty grouper). Since in the Galapagos the majority of fishers do not dedicate exclusively to one type of fishery throughout the year [61], this measure could also affect fisher's livelihoods and revenues during those months, as the time of peak spawning activity overlaps with the spiny lobster fishery seasonal ban (i.e., July to December; [83]). Finally, there is a necessity to establish new market-oriented incentives (e.g., improve value chains in the market) to reduce fishing pressure in bacalao and other related species of the fishery and to improve the effectiveness of existing co-management tools (e.g., active consultation) and communication and education to fishers, in order to facilitate bottom-up decision-making in fisheries management.

## 4. Conclusions and Lessons beyond This Study

Our results indicate that bacalao is a grouper that needs to be prioritized for urgent intervention in the Galapagos' handline fishery, given its limited resilience potential (as is also the case with many other related species), high market and cultural value, and interaction with gear and fishery spatial dynamics. Using PSA, we have also identified other related species at risk. Under the current state of the handline fishery, species like the leather bass and misty grouper might become threatened in the future. However, it is important to consider that our results are specific to this particular type of fishery situation and did not consider the susceptibility of these species to other fishing gear and threats (e.g., emergent threats as climate change and marine pollution). Moreover, when contrasting our results with the ones from other vulnerability assessments (i.e., IUCN Red List, FishBase indicators), we were able to obtain a more holistic view of vulnerability, and a better understanding of the linkages between fishing pressure and population trends.

The evaluation of hypothetical fisheries management scenarios using PSA, supported other studies (i.e., [40,76]) which suggested that bacalao would benefit from the addition of no-take zones in coastal

waters, especially if they protect key nursery habitats like mangroves, and the protection of spawning aggregation sites. This measure is already being considered since 2016, under the rezoning plan of the GMR [48,86]. However, as knowledge of the location of spawning aggregations sites is scarce, we believe that the protection of coastal waters using spatial closures might produce a short-term positive outcome for bacalao and other related species highly vulnerable to exploitation (e.g., misty grouper, leather bass), or already undergoing population decline (e.g., camotillo). In addition, we believe that these measures would be the most viable under the current socio-economic and political (and governance) context in Galapagos artisanal fisheries. Though, in order to guarantee the effectiveness of these measures, new directions are necessary to increase enforcement and compliance of fishers in the GMR. In addition, the implementation of parallel market-oriented incentives, and facilitation of co-management tools and education programs, would be necessary to reduce fishing pressure in bacalao and species affected by the handline fishery, and to increase participation of fishers in the bottom-up decision-making process, respectively.

Using the case study of bacalao and the Galapagos' handline fishery, we have demonstrated that PSA is a suitable method to estimate vulnerability in data-poor species and guide decision-making in multispecies fisheries. The application of this method is especially urgent in tropical areas, where fisheries data is limited, and fishing practices usually affect not only target, but accompanying species (or bycatch), as well as their habitats and communities. As PSA vulnerability indices reflect the risk of exploitation of a species to a specific type of fishery, we could identify those species that would need to be prioritized for intervention in the near future, given their highly vulnerable condition. This way, the results could be articulated within larger decision-support frameworks (e.g., ERAEF) and contribute towards EBFM objectives in a particular location. One of the advantages of PSA is that it could be implemented with little effort by local scientists and managers, as the nature of this method is semi-quantitative, and could rely on expert knowledge when assigning risk scores to a parameter. Additionally, the method could be adapted for the availability of information in a specific location, even including socio-economic variables within the susceptibility attributes.

Throughout this article, we have reviewed different interpretations of the concept of vulnerability in marine species impacted by fishing, and the implications for guiding decision-making in fisheries management. Although fisheries (specially small-scale) are of paramount importance for several sustainable development goals (SDG), in particular SDGs one and eight, which are related to poverty and economic growth, as well as SDGs two and three, which are about zero hunger and good health, it can also negatively influence the ecosystem (SDG 14, life below water) [101] by decreasing the population of many fish stocks; some having limited probability of recovery. In this regard, it becomes important to understand the local intrinsic and extrinsic factors that influence vulnerability in fish stocks in order to (i) identify species that may become threatened to exploitation in the near future, (ii) prioritize intervention accordingly, and (iii) implement suitable management measures. Informed decisions are important in fisheries management as they can enhance the occurrence of win-win scenarios for both marine fish conservation and the livelihoods and economies that depend on their exploitation.

**Supplementary Materials:** The following are available online at http://www.mdpi.com/2071-1050/12/17/6931/s1, Table S1. Formulas for the calculation of different indicators used in Productivity–Susceptibility Analysis. The calculations were taken from the Risk-Based Framework worksheet available from the Marine Stewardship Council website; Table S2. Susceptibility risk scores (for Productivity–Susceptibility Analysis) assigned to bacalao (*Mycteroperca olfax*) and related species (groupers and serranids) commonly caught in the Galapagos handline fishery, for each of the hypothetical management scenarios (S2 to S8) aiming to recover bacalao stock. Risk scores marked in pink represent changes from the state condition (S1), and scores marked in dark pink represent changes from S4.

**Author Contributions:** Conceptualization, J.F.P.-C.; methodology, J.F.P.-C.; formal analysis, J.F.P.-C., writing—original draft preparation, J.F.P.-C.; writing—review and editing, S.B., J.R.M.J., J.R.-G., J.R.B.-M., P.L.M.G.; supervision, J.R.B.-M., P.L.M.G.; project administration, P.L.M.G.; funding acquisition, P.L.M.G. All authors have read and agreed to the published version of the manuscript.

**Funding:** PhD studies of J.F.P.-C. were funded by the Special Research Fund (BOF) of Ghent University, Belgium and a top-up scholarship provided by the Escuela Superior Politécnica del Litoral (ESPOL), Ecuador. J.R.-G. was supported via a grant of the Gordon and Betty Moore Foundation.

**Acknowledgments:** We would like to thank the Charles Darwin Foundation team for providing us access to some unpublished literature from the Galapagos, principally theses and technical reports. Thanks also to Ramon Espinel Martínez and Julie Nieto, former and current Deans of Life Science Faculty at ESPOL-Ecuador, for their continuous support and diligence during the process of admission of J.F.P.-C. to his current PhD program at Ghent University—Belgium. Finally, we thank the Galapagos National Park Service for granting the research permit No. PC-41-20 to conduct this study. This publication is contribution number 2359 of the Charles Darwin Foundation for the Galapagos Islands.

**Conflicts of Interest:** The authors declare no conflict of interest. The funders had no role in the design of the study; in the collection, analyses, or interpretation of data; in the writing of the manuscript, or in the decision to publish the results.

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
