# Peer review of "Vulnerability and Decision-Making in Multispecies Fisheries: A Risk Assessment of Bacalao (Mycteroperca olfax) and Related Species in the Galapagos’ Handline Fishery"

_sustainability, doi:10.3390/su12176931_

Round 1

Reviewer 1 Report

Comments for Sustainability 887475 V1

This MS used Productivity-Susceptibility Analysis to prioritize the risk of an endemic grouper species -- bacalao (Mycteroperca olfax) and compared it with other 10 phylogenetically related species caught in the Galapagos’ handline fishery. Various management measures were also evaluated and the authors concluded that include additional no-take zones in the marine reserve, comprising key nursery habitats and spawning aggregation sites, would be necessary to promote stock recovery and benefit other related species. This MS made a very comprehensive review of the research of bacalao and the results and conclusion will provide very useful information for conservation of this species and its related species. This MS is well written and almost ready for publishing. However, there are some minor clarifications need to be made before it can be fully accepted.

In Table 1, the vulnerability score of misty grouper is “very high” from FishBase. However, “low risk” was found in this MS. As the maximum size is the largest among all 11 species examined in this study, the authors should provide some explanations.

Some of the life history parameters of bacalao are different from those in FishBase. For example, the maximum size is 120 cm in FishBase which is larger than 100 cm used in this study.

Table 2, please explain why use different life history parameters to determine the risk score for the parameter “selectivity”. Maximum size was used for low risk but size at first maturity was used for medium and high risk.

Lines 633-635, In addition to create new market-oriented incentives, communicating and educating fishermen to participate the bottom-up management decision making will be a key factor for ensure the sustainability of this species.

Please provide the calculation formulae for susceptibility index and PSA score in supplemental tables.

Author Response

Thank you for your comments. Please see the attachment for replies to R1.

Reviewer 2 Report

A very well written paper that describes a small-scale, multi-species fishery in the Galapagos, and the application of the relatively simplistic but useful PSA approach to examine the vulnerabilities of captured species to exploitation by this fishery. Whilst this method does not provide estimates of stock status for species, it is a valuable tool to use in data-poor situations to help prioritise resources available for monitoring and assessment. I particularly liked the testing of different management scenarios to explore their effectiveness in changing the susceptibility scores for species.

I think the paper could be improved by shifting some of the focus of the initial discussion of the PSA results (section 3.2.1) from comparisons with IUCN listings, which as has been stated in the introduction provide the global conservation status of species and is based on quite different attributes to the PSA, which is fishery-specific. It would be more appropriate to discuss the differences in, for example, the productivity scores between the PSA and the FishBase indicator, which are more comparable, and explain what is causing these.

Minor suggestions:

Lines 56-57 – Change 'small-scaled' to 'small-scale'

Lines 65-67 – extinction from exploitation is unlikely as it typically becomes un-economical to continue targeting stocks once they reach very low stock levels. Perhaps rephrase sentences in paragraph from ‘vulnerability to extinction’ to ‘vulnerability to exploitation’.

Line 125 – change ‘caught on’ to ‘caught by’

Line 133 – Rephrase ‘AFS is available for most marine fish’ as it is not an appropriate use of the acronym, which refers to the society. Also not clear how widely applied this approach is outside of America, might need to be reworded?

Lines 233-234 – does the ‘greater than 2 nautical miles’ refer to the distance from the shore?

Line 240 – change ‘with’ to ‘within’ or ‘by’

Lines 267-269 – this example makes it sound like the areal overlap with the fishery would be less if a species range is more restricted, which may not be the case (depends on extent of the fishery). This category should be scored based on the overlap of the fishery with the species range (not the opposite way around), so if the fishing effort occurs over more than 30% of the stock distribution, it would be considered a high risk.

Line 318 – change ‘recollection’ to ‘collection’

Line 324 – add ‘species’ after iconic

Line 393 – add ‘of’ after ‘lack’

Line 553 – rather than ‘revert the overfished status’ it would be more appropriate to refer to it as reducing the vulnerability of the species by decreasing their susceptibility to the fishery.

Line 556 – Not specifically marine reserves, but rather spatial closures, where no-take zones in a marine reserve is an example.

Lines 595-596 – sentence beginning with ‘after implementation’ does not make sense – reword.

Line 607 – change ‘politic’ to ‘political’

Section 3.2.2 – when referring to management scenarios in the discussion, state what they were rather than S2, S3 etc. so reader doesn’t have to keep going back to check.

Throughout the manuscript – the acronym AFS is not appropriately used in a few instances as it refers to the name of the American Fisheries Society (as defined in the introduction) and should not be used to refer to the DPS concept used to generate the FishBase resilience indicator.

Author Response

Thank you for your comments. Please see the attachment for replies to R2.
